# Development and Characterization of an Ex Vivo Testing Platform for Evaluating Automated Central Vascular Access Device Performance

**DOI:** 10.3390/jpm12081287

**Published:** 2022-08-05

**Authors:** Emily N. Boice, David Berard, Sofia I. Hernandez Torres, Guy Avital, Eric J. Snider

**Affiliations:** 1U.S. Army Institute of Surgical Research, JBSA Fort Sam Houston, San Antonio, TX 78234, USA; 2Trauma & Combat Medicine Branch, Surgeon General’s Headquarters, Israel Defense Forces, Ramat-Gan 52620, Israel; 3Division of Anesthesia, Intensive Care & Pain Management, Tel-Aviv Sourasky Medical Center, Tel-Aviv 64239, Israel

**Keywords:** model development, automation, ex vivo, porcine, femoral, vascular access device, medical devices

## Abstract

Access to the central vasculature is critical for hemodynamic monitoring and for delivery of life-saving therapeutics during emergency medicine and battlefield trauma situations but requires skill often unavailable in austere environments. Automated central vascular access devices (ACVADs) using ultrasound and robotics are being developed. Here, we present an ex vivo lower-body porcine model as a testing platform for evaluation of vascular devices and compare its features to commercially available platforms. While the commercially available trainers were simpler to set-up and use, the scope of their utility was limited as they were unable to provide realistic anatomic, physiologic, and sonographic properties that were provided by the ex vivo model. However, the ex vivo model was more cumbersome to set-up and use. Overall, both have a place in the development and evaluation pipeline for ACVADs before testing on live animals, thus accelerating product development and translation.

## 1. Introduction

Hemorrhage is the leading cause of preventable death for civilian [1] and military trauma casualties [2]. Blood product resuscitation is vital for the survival of bleeding casualties [3] and necessitates expeditious securement of vascular access [4]. Adequate vascular access is required for the management of trauma to provide resuscitation fluids, as well as a route for medications and intravenous contrast for diagnostic procedures [5]. Furthermore, central vascular access is vital for the performance of additional life-saving procedures, such as Resuscitative Balloon Occlusion of the Aorta (REBOA) [6,7] and Extracorporeal Machine Oxygenation (ECMO) [8]. Central vascular access requires skilled personnel, especially in cases of profound shock [9], who might be available in trauma units and emergency departments but are unlikely to be available during combat casualty care or remote emergency medicine scenarios.

To address the expertise gaps, research, particularly prioritized by the Department of Defense [10], is being conducted to develop interventions, such as automated central vascular access devices (ACVADs). Devices such as the VuPath, by Crystalline Medical [11], and AI-GUIDE (Artificial Intelligence Guided Ultrasound Interventional Device) by the Massachusetts Institute of Technology Lincoln Laboratory [12] are in various phases of development for the semi-autonomous ultrasound guidance of needle placement. These devices, as well as many others in development by academic and industry partners, utilize ultrasound imaging to identify the central artery or vein of interest, followed by needle alignment for manual or automated needle insertion. These designs are typically meant to be compact and even handheld, to allow use in remote and austere conditions, such as battalion aid stations and mobile evacuation platforms (such as ambulances, helicopters, boats, and armored personnel carriers), where the expertise for central vascular access might also be lacking. This is an essential step in the effort to push advanced resuscitation technologies such as REBOA and ECLS forward toward the point of injury.

With devices in development, it is imperative to standardize a testing methodology for evaluating the performance of devices through multiple variations of anatomy and physiology. Testing platforms can range from commercially available simulators to live animal models. While commercially available benchtop platforms are easy to set-up, they may not offer realistic anatomical, physiological, or sonographic properties. Previous work from our lab developed a modular tissue phantom to address some of these limitations [13]. This testbed is easy to fabricate and use but lacks anatomical complexities. Live animal testing could solve portions of this problem, but its cost in animal suffering, time, and resources hinders its utility for individual device troubleshooting and comparing performance across a wide range of devices. Here, we develop an ex vivo euthanized animal model and compare it with commercially available platforms for measuring performance criteria key for evaluation of vascular access devices. The ultimate goal of this work is to standardize a military-relevant testing pipeline for ACVADs.

## 2. Materials and Methods

### 2.1. Evaluated ACVAD Performance Metrics

A matrix of performance metrics was compiled to evaluate the capabilities of automated vascular access devices. Metrics were selected from an engineering perspective for developing vascular access devices and comparing their performance to others. These include successful insertion, accuracy of the final needle tip position with respect to the vessel walls, and angle of insertion. These metrics must be measured in states mimicking normovolemia to severe hypovolemia as the device needs to be suitable for performance in both instances. Further, anatomical variability in vessel location and size is critical in a testing platform to confirm whether a device’s performance has not been overfitted without accounting for subject variability. Another important aspect for the initial design of a vascular access device is the trauma caused by needle insertion to the vessel or surrounding tissue. This may be caused by the device being too bulky to hold steady, repeated attempts to insert the needle, or the speed of vascular cannulation not being tuned appropriately. Additional criteria were also identified for comparing and evaluating vascular access devices, but they are device criteria as opposed to insertion performance criteria highlighted above. Device criteria not considered at this time are summarized in Appendix A.

Two levels of vascular models were evaluated for their ability to measure the described vascular access performance metrics: (1) commercially available ultrasound trainers and (2) a developed ex vivo lower-body porcine model.

### 2.2. Commercially Available Ultrasound Trainers

Two commercially available ultrasound trainers were used: Generation II Femoral Vascular Access and Regional Anesthesia Ultrasound Training Model (CAE Blue Phantom, Sarasota, FL, USA) and Regional Anesthesia Femoral Trainer with SmarTissue (Simulab Corporation, Seattle, WA, USA). Commercially available ultrasound images [14,15] and previous research [12,16,17,18,19] with these trainers have shown that they contain clearly identifiable vascular structures in ultrasound compliant tissue to mimic human anatomy and were selected for use in this work. Ultrasound compliant fluid provided by respective manufacturers for each trainer were added to arteries and veins and the removal of air bubbles was ensured. Hand pumps provided with each vascular access trainer allowed for pulsating the arterial lines to mimic pulsatile flow into the closed system.

### 2.3. Lower-Body Ex Vivo Porcine Model

Euthanized swine tissue was procured from commercial vendor (Animal Technologies, Tyler, TX, USA) or from an unrelated Institutional Animal Care and Use Committee approved animal protocol. The ex vivo model was developed, using an exsanguinated, eviscerated lumbar-to-shin section (Figure 1) that was prepared approximately 24 h prior and kept on ice. 8Fr feeding tubes (Covidien, Mansfield, MA, USA) were advanced through the vessels from proximal (vena cava and aorta) to distal (superficial femoral vessels) to reopen the collapsed vessels and support their localization. The gracilis and sartorius layers of muscle were bluntly dissected through the fascial planes and divided near their origins to create a flap and allow full access to the vessels while preserving the femoral sheath. 14G IV catheters (MedOfficeDirect, Naples, FL, USA) with the needles pre-removed were used to cannulate the superficial femoral vessels and locked into position with Perma-Hand Silk Ligatures (Ethicon, Raritan, NJ, USA). 8Fr PCI introducers (Argon Medical Devices, Athens, TX, USA) were used to cannulate the external iliac artery and vein through the aorta and vena cava, respectively, and sutured into place proximally. The external iliac vessels were ligated around the introducers’ tip to create a seal. The cannulas in the superficial femoral vessels were connected to create an arterio-venous shunt loop consisting of a 3-way stopcock (Cole-Parmer, Vernon Hills, IL, USA) and required tubing and luer fittings to connect to femoral vessels. Water was added to the reservoir (Figure 1, labeled Res) and the peristaltic pump (Cole-Parmer, Vernon Hills, IL, USA) rate was adjusted to prime the lines, vessels, and shunt, and to remove air. The shunt was partially clamped, simulating arteriolar and capillary pressure gradients (Figure 1, Arterio-venous shunt). Proximal canulations of the external iliac vessels were used to connect the ex vivo porcine model to a combined hydrostatic IV bag reservoir (Baxter, Deerfield, IL, USA) in a pressure cuff (Infusable^®^, Vital Signs Inc., Barnham, UK) (Figure 1A, IV) and a peristaltic pump-based pseudo-perfusion system (Figure 1, Pump). This setup means tissue perfusion is not being performed, but rather an antegrade flow in the major vessels, allowing for separate control of the simulated diastolic and pulse pressures (Figure 1). Venous pressure and, therefore, diameter were controlled by adjustment of drainage height (Figure 1, Venous Drain). After the flow was initiated, leaking vessel branches were identified and ligated. Pressure transducers were placed immediate to the inflow and outflow of the proximal cannulation sites through 3-way stopcocks (Figure 1, PT_V_ and PT_A_) and monitored using a Delta XL Patient Monitor (Drager, Lubeck, Germany). The muscle flap was repositioned and fixated in its edges using towel holders to restore anatomic position. Vessel internal diameters were sonographically measured at the level of the femoral crease. Peristaltic pump flow rates, pressure on the IV bag, the shunt clamp, and venous drain height were adjusted to produce hypovolemic-like vessel pressures and diameters.

### 2.4. Ultrasound Imaging and Vascular Access

For all the experimental platforms, imaging was performed with HF50 and L25 probes (Fujifilm, Bothell, WA, USA) with a Sonosite Edge II Ultrasound (Fujifilm, Bothell, WA, USA) system. Aquasonic Clear Ultrasound Gel (Parker Laboratories, Inc., Fairfield, NJ, USA) was applied and an ultrasound probe was positioned at the model’s surface near the inguinal crease. B-mode images and video clips were captured with both longitudinal and transverse probe placement at varying depths depending on platforms, to achieve a clear view of the vessels. Analogous to the benchtop platforms, lower-body ex vivo porcine tissue ultrasound images were obtained at varying depths and angles in the femoral triangle. These images were captured at normo- and hypovolemic simulating states in the lower-body ex vivo porcine tissue model. Images were adjusted for brightness to better illustrate needle positioning.

For vascular access, transverse, out-of-plane (OOP) ultrasound images of the vein and artery were obtained with the target vessel near the center of the image. A 2.5 in, 18-gauge needle (Medline, Northfield, IL, USA) was manually inserted in an OOP approach at an approximate angle of 45° to the surface and visualized in real-time during its advance through superficial and deep fasciae into the target vessel. Successful cannulation was confirmed by aspiration of fluid into the syringe. Following insertion, the needle was stabilized, and in-plane (IP) images were obtained to demonstrate needle location in the lumen in the vertical plane.

## 3. Results

### 3.1. Commercial Test Results

For developing a test platform for vascular access devices, we first evaluated commercially available trainers that are ultrasound compliant. Specifically, the Blue Phantom Gen II Vascular Access training model and Simulab Regional Anesthesia Femoral Trainer with SmarTissue were evaluated. Manual vascular access was performed for both arterial and venous access to determine which performance metrics could be evaluated with each model.

For the Blue Phantom model, the phantom had variable artery and vein depth at locations across the tissue phantom leg, allowing for more variable vascular access evaluation. In addition, the tissue was ultrasound compliant, which made tracking the needle insertion at deeper depths obvious. Needle insertion into the artery was tracked in both out-of-plane (Figure 2A) and in-plane (Figure 2B) views, which allowed for the evaluation of needle position and angle of entry. Performance metrics were capable of being measured for needle distance to the vessel (Figure 2C) and angle of insertion (Figure 2D). Representative measurements are shown for the venous side. The Blue Phantom model was not equipped for flow, and pulsatile vessel expansion was only possible with a hand pump.

The Simulab model had more superficial vessel anatomical locations, but the model was more hyperechoic, making visualizing the vessels more challenging, and thus needle insertion was more difficult to track. After adjusting the brightness of the images in post-processing, the needle was evident in both out-of-plane (Figure 3A) and in-plane (Figure 3B) views, allowing for measurements of the needle position and angle of entry. The same was true for both arterial (Figure 3A,B) and venous procedures (Figure 3C,D). However, similarly to the Blue Phantom trainer, the model was not designed for flow scenarios. A hand pump allowed for trying to mimic pulsatile expansion of the artery, but without real flow or control on vessel diameter. Lastly, one identified criterion for vascular access evaluation was focused on needle insertion-related damage to the vessel. Unfortunately, both commercial models cannot measure this as the vessels are empty chambers within the silicone as opposed to containing a vessel wall that can be inspected after the vascular access procedure.

### 3.2. Lower-Body Ex Vivo Porcine Model Results

Next, the ex vivo model was used to simulate two physiologic profiles—normal and profound hypovolemic shock. As expected, some variability was seen between repetitions of the experiment on different tissue specimens (Figure 4). The normal physiological profile for the first tissue specimen was characterized by relevant porcine arterial pressure (110/80 mmHg) and venous pressures (10 mmHg), and porcine arterial diameter and venous diameters (6.5 mm and 8.7 mm) comparable to previously available porcine ultrasound images. Similarly, when the flow was reduced and the hydrostatic reservoir lowered, the tissue specimen mimicked a profound hypovolemic shock profile (Figure 4A). Venous diameter represents a mean of the short and long dimensions of the vein, which tends to flatten with decreasing volume. This was very apparent in the comparison of the venous diameters in normo- (Figure 4B) and hypovolemic conditions (Figure 4C).

Needle insertions were successfully performed in all profiles on the ex vivo model. US images of arterial and venous cannulations were recorded OOP during insertion (Figure 5A,C). IP was recorded immediately after with the needle stabilized (Figure 5B,D). With use of the porcine tissue specimens, the vessels’ surrounding tissue was realistically heterogenous with the muscle, fascia, and nerve tissue apparent, as well as the immediate adjacency between the vessels, unlike the synthetic models (Figure 5C). The use of different tissue specimens also produced subject variability. Thanks to the pseudo-perfusion system connected to the lower-body ex vivo porcine model, and the realistic physical properties of the vessels, measurement of vascular access needle placement and insertion angle could also be evaluated in states simulating normo- and hypovolemic scenarios (data not shown). However, the complexity involved in removing a particular vessel after vascular access for inspection of vessel damage was not practical in the model and was identified as not being suitable for performance metrics requiring that.

## 4. Discussion

Central vascular access is critical for monitoring the patient’s condition accurately and providing live-saving therapeutic interventions in emergency medicine or combat casualty care, but vascular access remains challenging to perform even by trained medical personnel. To facilitate the process, automated vascular access devices are in development by academic and industry institutions employing different methodologies and testing procedures. Thus, a standardized testing scheme is needed in order to provide a relevant platform for troubleshooting product development and to objectively compare performance between potential devices. Here, we developed an ex vivo animal model and compared its performance to two commercially available trainers to assess the role of each model for assessing key performance criteria for central vascular access devices.

Performance criteria were identified for general device performance as well as criteria-associated patient safety. Some of the parameters, while by themselves are of minor importance in clinical practice, such as the final needle tip position in relation to the vessel walls, might indirectly represent the accuracy and safety margins of the device. The importance of each parameter depends on the intended use of the device. For instance, devices intended to achieve venous access for high-flow fluid administration should perform well under conditions simulating hypovolemia. A methodological assessment of clinical needs, such as an end user survey study, may be used to assign the relevant weight to each parameter when scoring a device in the context of a specific purpose. One limitation with the current metrics used in this work was that many potential evaluation metrics are device-specific, such as whether the device has artificial intelligence to know when the vessel has been missed or ease of use by the operator. These additional criteria are summarized in Appendix A.

While the test platforms described here may be suitable for these additional metrics, they were not measured at this time in the absence of an actual ACVAD. The test platform developed in this work included methods up to but not including live animal testing in order to develop a cost-effective, widely available platform that can readily be used when developing and testing a vascular access device. Each model had their advantages and disadvantages (Table 1). The commercial trainers allowed for off-the-shelf evaluation of needle insertion into an artery and vein and measurement of overall performance metrics. However, without the capability to be connected to a pseudo-perfusion system, and without an actual vessel wall, both were unable to mimic hypovolemic conditions or different vessel sizes, accommodate doppler-reliant devices, or evaluate the damage to the vessel after insertion. However, an exhaustive evaluation of all vascular trainers was not performed, instead focusing on prominent, common options. However, it is unlikely that any model currently available can address the many shortcomings identified for vascular access testing. In contrast, the lower-body ex vivo porcine model allowed for evaluation of most testing criteria. Normo- and hypovolemic mimicking states could be created by adjusting flow rates of the system to reduce overall pressure, and swine tissue was obviously more sonographically realistic than commercial trainers, challenging the AI-driven algorithms guiding various ACVADs. While it was not evaluated, doppler compliance may be possible with different fluid types or colloid suspensions, and damage to the vessel may be evaluated with a much more extensive tissue dissection procedure. Overall, the euthanized tissue is more appropriate for physiological evaluation of vascular access devices, but it is cumbersome to set-up, with a significant resource and training burden to do so properly, although significantly less than live animal experiments.

In summary, we recommend the commercial trainers be used for initial product design troubleshooting as they are easy to use, but more realistic benchtop evaluation with high-fidelity sonographic appearance, subject variability in vessels, and hypovolemic use cases can only be achieved with the ex vivo testing platform. Given the challenging setup process for this model, there is a need for more robust, realistic vascular access trainers to better mimic those properties to simplify and aid in the product development prior to live animal testing, in order to reduce animal suffering and costs. This will accelerate product development as an advanced vascular access trainer can be much more streamlined compared to animal work, allowing for quicker troubleshooting and comparison to other design approaches. In addition to this recommendation, the next steps for this work will be to develop a standardized live animal model that can be used for normo- and hypovolemic device evaluation and add this work as the next step in the testing pipeline proposed here. Incorporating this missing piece will encompass the entire pre-clinical testing methodology for ACVAD evaluation and accelerate product translation for use in providing life-saving therapies during emergency medicine and combat casualty care.

## Figures and Tables

**Figure 1 jpm-12-01287-f001:**
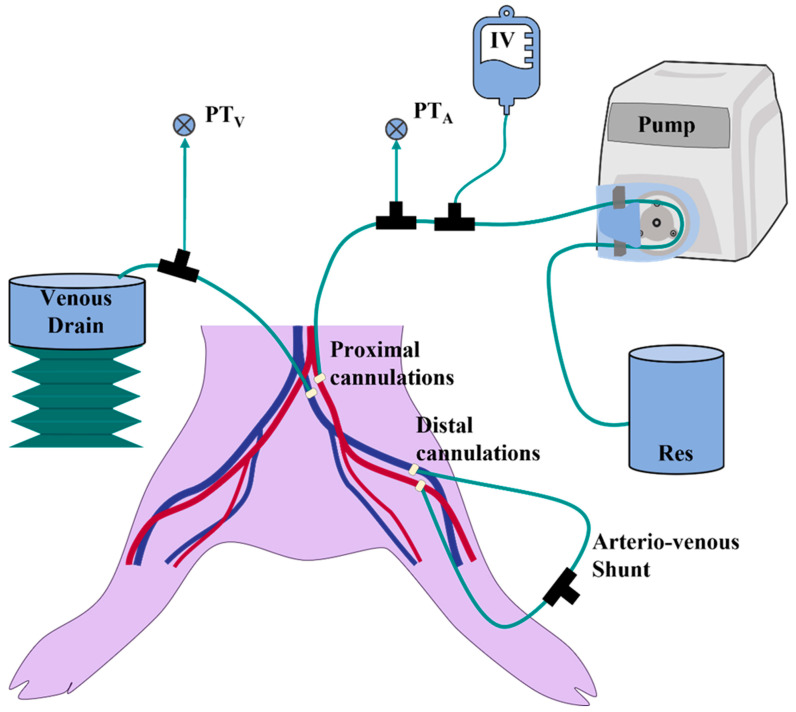
Lower-body Ex Vivo Porcine Model. Illustration of the cannulations sites into the femoral region of the porcine legs and integration into the fluidics system with a water reservoir (Res), peristaltic pump (Pump), adjustable hydrostatic reservoir (IV), arterial pressure transducer (PT_A_), venous pressure transducer (PT_V_), cannulation sites, arterio-venous shunt, and venous drain on adjustable jack (Venous Drain).

**Figure 2 jpm-12-01287-f002:**
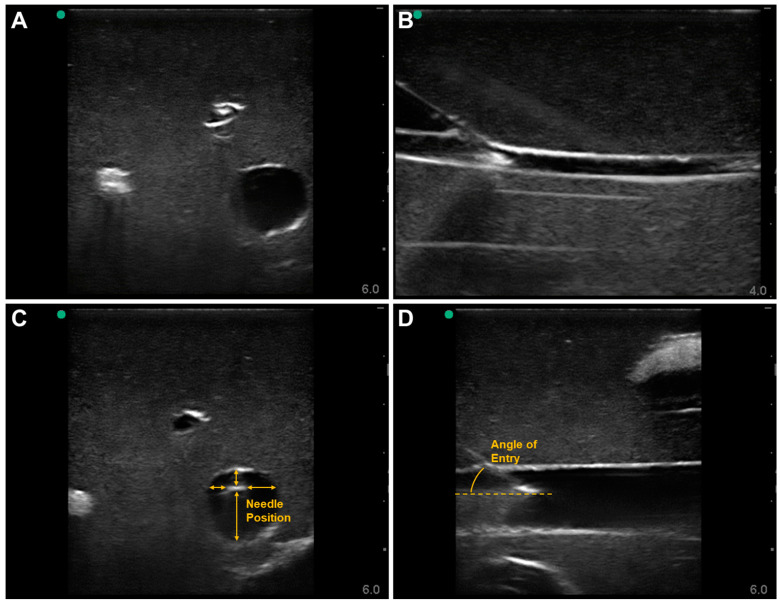
Commercially available Blue Phantom Gen II Vascular Access model with out-of-plane (**A**) and in-plane (**B**) views of arterial needle insertion and out-of-plane (**C**) and in-plane (**D**) views of venous needle insertion. Measurements were overlayed in the ultrasound images (**C**,**D**) for how key performance metrics can be collected from the ultrasound frames.

**Figure 3 jpm-12-01287-f003:**
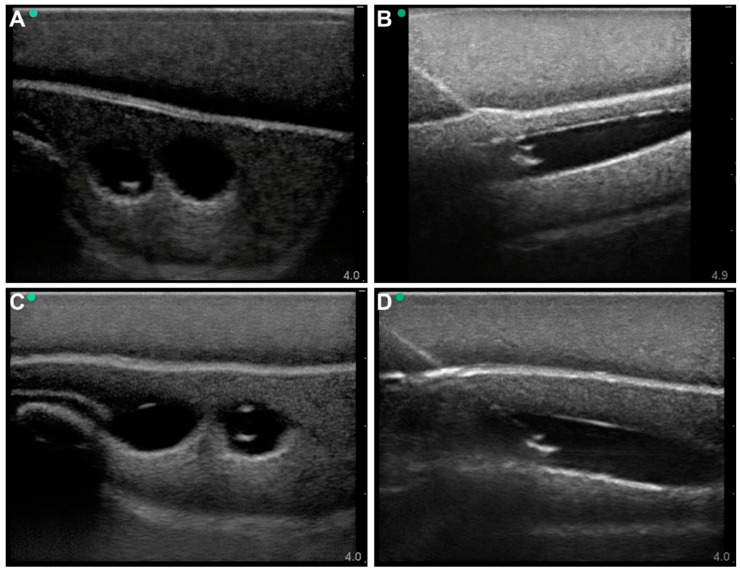
Commercially Available Simulab Regional Anesthesia Femoral Trainer with SmarTissue System Vascular Access Model. Out-of-plane (**A**) and in-plane (**B**) views of arterial needle insertion and out-of-plane (**C**) and in-plane (**D**) views of venous needle insertion.

**Figure 4 jpm-12-01287-f004:**
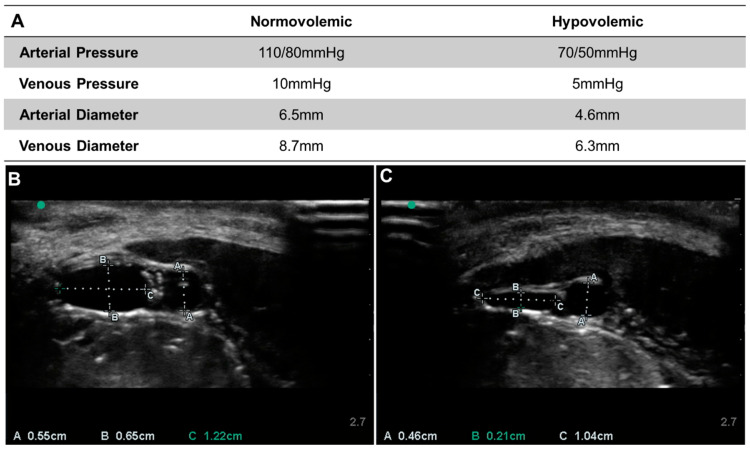
Dimensions for the Lower-body Ex Vivo Porcine Model. (**A**) Normovolemic and Hypovolemic measurements for pressure and diameters for the first ex vivo porcine specimen. (**B**) Normovolemic and (**C**) Hypovolemic ultrasound images and dimension overlays for an additional ex vivo model specimen.

**Figure 5 jpm-12-01287-f005:**
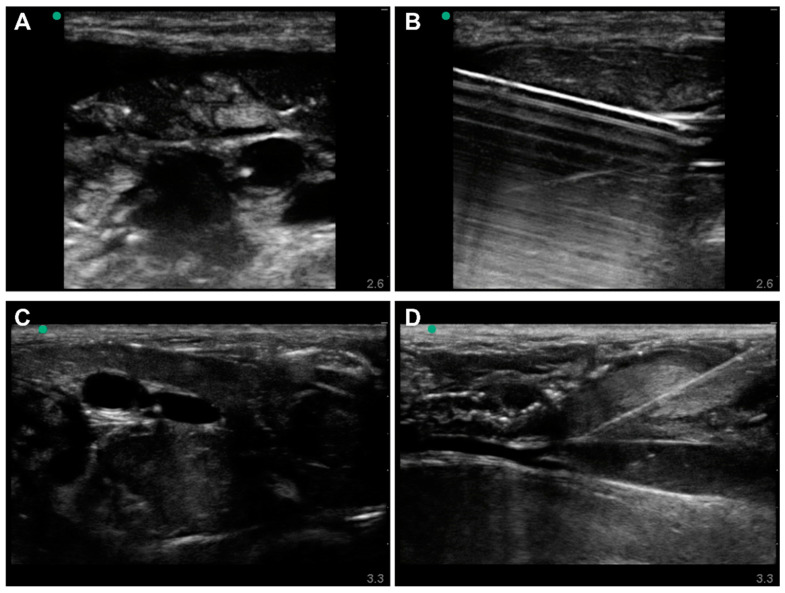
Lower-body Ex Vivo Porcine Model under Normovolemic Conditions. Out-of-plane (**A**) and in-plane (**B**) views of arterial needle insertion. Out-of-plane (**C**) and in-plane (**D**) views of venous needle insertion.

**Table 1 jpm-12-01287-t001:** Summary of each model platform’s ability for measuring key performance metrics for vascular access devices. Checkmark indicates model could successfully perform. ** was not evaluated, but likely can be accommodated.

Model Features for Evaluating Vascular Access	Blue Phantom Gen II Femoral Ultrasound Trainer	Simulab Regional Anesthesia Femoral Trainer	Lower-Body Ex Vivo Porcine Model
Can measure needle bore tip relative to all vessel walls	✓	✓	✓
Angle of needle to surface can be visualized	✓	✓	✓
Needle flashback for successful insertion into vessel	✓	✓	✓
Subject vessel depth variability	✓		✓
Subject distance between vessels variability	✓		✓
Subject vessel diameter variability			✓
Variety of volume states			✓
Physiological Ultrasound Complexity			✓
Doppler Compliant			**
Extent of vessel damage from needle insertion can be measured			

## Data Availability

The datasets generated during and/or analyzed during the current study are available from the corresponding author upon reasonable request.

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
