# Peer review of "Development and Characterization of an Ex Vivo Testing Platform for Evaluating Automated Central Vascular Access Device Performance"

_jpm, 2022, doi:10.3390/jpm12081287_

Round 1
Reviewer 1 Report
This is a very meaningful manuscript; the study indicates that an ex vivo lower-body porcine model as a testing platform for evaluation of vascular devices and compare its features to commercially available platforms. The manuscript is well-written and explanatory. Method and Results are well-defined. The language is understandable with no grammatical and syntaxerrors. The references are up-to-date. I recommend the acception of this manuscript for publication.
Reviewer 2 Report
Dear authors, thank you for the opportunity to review this manuscript. The authors present a simulation-setup using an ex-vivo porcine model. The context/rationale for this is the presumed requirements for accurate/realistic training/evaluation of future automated VAD insertion tools - this is novel and exciting.
The manuscript reads well and is balanced throughout. I (and others as well I presume) like the level of innovation and exploration of new techniques.
However, I am struggeling to see where/when the fully automated devices will be used in clinical practice. Even though this is not the primary aim of the paper, the authors make a point of automated VAD insertion /the use of AI in future combat medicine. As a reader, I would be hear the authors thoughts on this - is it applicable during transport? In the resus area? In the OR? This is important for context and deserves a line or two in the introduction or perhaps even more appropriate in the discussion.
